# Food Neophobia in Children with Autistic Spectrum Disorder (ASD): A Nationwide Study in Brazil

**DOI:** 10.3390/children9121907

**Published:** 2022-12-06

**Authors:** Priscila Claudino de Almeida, Renata Puppin Zandonadi, Eduardo Yoshio Nakano, Ivana Aragão Lira Vasconcelos, Raquel Braz Assunção Botelho

**Affiliations:** 1Graduate Program in Human Nutrition, University of Brasília, Brasília 70910-900, Brazil; 2Department of Nutrition, University of Brasília, Brasília 70910-900, Brazil; 3Department of Statistics, University of Brasília, Brasília 70910-900, Brazil

**Keywords:** food neophobia, autism, child

## Abstract

Food neophobia (FN) is common among children with autistic spectrum disorder (ASD), potentially impairing their health and diet quality. This study aimed to investigate and classify the prevalence of FN among 4-to-11-year-old Brazilian children with ASD. This descriptive cross-sectional study was performed by applying online a validated instrument to identify FN in Brazilian children with ASD through their caregivers’ responses for a national prevalence of FN in this group. The final sample included 593 children with ASD, 80.1% of boys, with a mean age of 6.72 ± 2.31 years, and 83% having only ASD. Almost 75% (*n* = 436) of the children with ASD had high food neophobia scores. The fruit neophobia domain had the lowest prevalence of high neophobia (63.7%). No significant difference in FN (total, fruit, and vegetable domains) was found, considering gender and age. There was no statistical difference in FN (all domains) by the number of residents in the same household, income, or the caregivers’ educational level. FN did not decrease in older children with ASD. FN is a more complex problem, requiring a multidisciplinary trained team to face the problem.

## 1. Introduction

Autistic spectrum disorder (ASD) is a heterogeneous neurodevelopmental disability that may lead to persistent deficits in social communication and interaction and behavioral challenges. It includes restricted and repetitive patterns of behavior, interests, or activities [1,2]. The Centers for Disease Control and Prevention (CDC) estimates the prevalence of ASD as 1 in 44 children and four times more frequent in boys than in girls [3].

The manifestations of ASD vary according to age, developmental level, and severity—hence the term “spectrum”. Typical behaviors are persistence on the same thing; an inflexible routine; monotonous eating habits; resistance to change; extreme reactions regarding food taste, smell, texture, or appearance; and excessive dietary restrictions due to hyperreactivity or hyporeactivity to sensory stimuli [1,2]. Considering these aspects, ASD has been associated with eating-related atypicality, such as food neophobia (FN). FN is characterized by the difficulty of trying novel foods that can potentially impair diet quality and health [4,5]. FN is influenced by heredity [6,7], prenatal experiences [8], parental influence on eating habits [9,10], parental pressure for the child to eat [9], parental affectivity during meals [9], child anxiety [9,11], sensory preferences [8], and an innate preference for some flavors [9]. FN has been investigated in several scenarios, such as in children [9], twins [6], adolescents [12], adult twins [13], adults [14], pregnant women [15], the elderly [16], and patients with celiac disease [17]; for specific foods [18,19]; and even during the COVID-19 pandemic [20]. The FN peak is in childhood, and there is a tendency of stabilization in adulthood [5,8,21,22,23]. All aspects related to food refusal may be worse in ASD due to its repetitive behavior and the reluctance to accept new issues or changes in routines [24,25,26].

Environmental factors, such as parents’ behaviors, mainly affect the food choices of a child with ASD and may encourage them to try a more diversified and healthier diet [24]. A high FN level is frequently related to low consumption of fruits and vegetables (the most common food neophobia in children) [21,27] and a higher intake of sweets and snacks [28,29,30]. Several studies have been performed on the nutritional interventions and eating disorders of children with ASD [24,25,26,31] but few on FN in this group [4,32,33]. No study has either evaluated the FN prevalence and classification in children with ASD or whether it occurs only during the peak that is up to 5 years and then decreases [9]. The hypothesis is that there is a high prevalence of FN in children with ASD due to their stereotypes, rigidity, and difficulties in social environments.

Data on the prevalence and classification of FN in children with ASD can help health professionals and parents with appropriate mealtime behavior and healthier eating habits in children with ASD. Therefore, this study aimed to investigate and classify the prevalence of FN among 4-to-11-year-old Brazilian children with ASD.

## 2. Materials and Methods

This research was a descriptive cross-sectional study performed by applying a previously validated instrument [34] in a sample of caregivers of children with ASD to evaluate and characterize the prevalence of FN in 4-to-11-year-old Brazilian children with ASD. According to the Declaration of Helsinki guidelines, the study was approved by the Health Sciences Ethics Committee, University of Brasilia (no. 4.407.816, November 2020), updated in 2022 (no. 5.438.498).

### 2.1. Participants

The sample comprised caregivers of children with ASD from Brazil (diagnosed by a physician using DSM-5) who consented to participate in the study and knew about the children’s eating habits (often following their meals). The caregiver was considered the person who cared for the child, such as the mother, father, a grandparent, brother, sister, uncle, or cousin, or a person who cares for the child with no family relationship (Appendix A). Researchers excluded incomplete questionnaires.

Only one caregiver could answer the instrument, as informed by the researchers. Those with more than one child with ASD who met the inclusion criteria could answer the questionnaire once for each child. The snowball sampling technique was used for recruitment, with convenience non-probability sampling. The snowball sampling technique uses reference networks, so it is appropriate for research with groups that are difficult to access or even when dealing with more private topics. This study used public and private education centers as reference networks, such as schools [35]; groups of children with ASD and parents of children with ASD, such as the Moviment Pride Autism Brasil [36]; national and state autism groups; national clinical contacts, such as pediatricians, psychologists, psychiatrists, and psychopedagogues and teachers who care for autistic people; regional councils of nutritionists [37,38]; and universities, university centers, and colleges in their projects with the community (extension projects) that assisted children with ASD. In addition, we used disclosures on the internet in social media, such as Instagram^®^ [39], Facebook^®^ [40], and Twitter^®^ profiles related to the studied group.

### 2.2. Instrument Application

The FN validated instrument [34] was spread using the online platform Google Forms^®^ via social networks (such as Facebook^®^, Twitter^®^, and Instagram^®^), messaging apps, and email from November 2020 to November 2021.

Sociodemographics related to the respondent caregiver and child data were collected in the first part of the instrument. The caregiver’s variables were age, gender, degree of kinship, marital status, educational level, and family income. Children’s profiles in the study consisted of the following variables: gender, age, diagnoses, Brazilian region of living, housing area, and the number of people living in the same house.

Caregivers answered the instrument themselves. The FN instrument presented three domains: general, fruit, and vegetable neophobia. A better analysis of the scores was possible because each domain had a homogenous number of items [34], allowing the assessment of FN in each domain or in general.

### 2.3. Statistical Analysis

Children with ASD were grouped by age into two categories (4–7 and 8–11 years) following a previous search [41] that mentioned that the cut-off point of FN intervention should start before 8 years of age. The children’s gender (male or female) was evaluated separately.

For accounting and classification, the response options were inverted on the scale, and the responses were scored from 0 to 4. The total score (25 items: general domain with 9 items, fruit domain with 8 items, and vegetable domain with 8 items) ranged from 0 to 100. Higher values indicated a higher FN of the evaluated child. Classification followed three levels of FN (low, medium, and high). Next, domain score cut-off points were up to 13 points (low), from 14 to 21 points (moderate), and 22 points or more (high). Moreover, the total score cut-off points were up to 40 points (low), 41 to 65 points (moderate), and 66 points or more (high).

The FN scores’ quantitative variables were determined using means and standard deviations. For categorical variables, frequencies were used. The FN scores for gender and age were compared using the independent Student’s t-test. The Kolmogorov–Smirnov (with Lilliefors adjustment) test verified the normality of scores. Pearson’s chi-square test was performed to compare the neophobia levels by gender, age, educational level of caregivers, number of residents in the same house, and family income. Two-tailed hypotheses and a significance level of 5% were considered for all tests. The Google Forms^®^ platform data were extracted and analyzed using SPSS^®^ 20.0 software.

## 3. Results

Of 742 participants who accessed the questionnaire, 593 constituted the final sample (Figure 1). Seventy-one children excluded from the sample were without ASD but had other conditions, such as Down syndrome, food intolerance, and allergies or had no diagnosed condition. Children with a confirmed diagnosis of ASD could have, in this study, other coexisting conditions beyond ASD.

Caregivers were mostly female (*n* = 565; 95.3%), in a stable relationship (*n* = 411; 69.3%), and mothers (*n* = 544; 91.7%) and had a high school degree (*n* = 163; 27.5%). The monthly family income was low; 57.5% received up to 3 minimum wages (MW), with the highest frequency up to 1 MW (*n* = 128; 21.6%) and up to 2 MW (*n* = 111; 18.7%). Some participants had no income (*n* = 32; 5.4%). During data collection, the MW was about USD 190 (conversion rate of USD 1.00 to BRL 5.52); see Appendix A.

Children were mostly boys (475; 80.1%) and living in urban areas (560; 94.4%), with a mean age of 6.72 ± 2.31 years (Appendix A) and a maximum of 10 people living in the same house (3.77 ± 1.14). Most children had only an ASD diagnosis (*n* = 492; 83%). The remaining also presented other medical diagnoses, such as food allergies/intolerance and Down syndrome (Appendix A).

Considering the total FN score, almost 75% (*n* = 436) of the children with ASD presented high scores for FN. The prevalence of low and high FN among children with ASD was 8.4% and 73.9%, respectively (Table 1). The FN domain for fruits had the lowest prevalence of high neophobia (63.7%).

There was no significant difference in FN (total, fruit, and vegetable domains) by gender and age (Table 2).

Results did not show a significant difference in FN (all domains) by the number of residents in the same household up to 3 people compared to 4 or more (*p* = 0.760; *p* = 0.538; *p* = 0.628; *p* = 0.698). There was no significant difference in FN by income when comparing up to 4 MW, 5 to 9 MW, and more than 10 MW (*p* = 0.178; *p* = 0.939; *p* = 0.403; *p* = 0.629). There was also not a significant difference in FN by the caregivers’ educational level (*p* = 0.184; *p* = 0.872; *p* = 0.196; *p* = 0.552); see Table 3.

## 4. Discussion

This study is the first to assess FN in Brazilian children with ASD using a validated questionnaire, bringing data from a large sample of children with ASD. It is crucial since FN can impair the nutritional status and health of children with ASD because they tend to experience a monotonous diet. Comprehending the factors that influence FN in children with ASD may help them and their families to improve their diet quality and plan their meals. Most respondents were female (*n* = 565; 95.3%) and mothers (*n* = 544; 91.7%), similar to those found in a nationwide study on FN in Brazilian children in which 92.4% of respondents were female and 86% were mothers [42]. These data were expected since females and mothers tend to be more concerned about children’s health and participate more often in studies [43,44,45,46,47]. It is important to emphasize that a previous study using the same FN instrument compared responses from parents (mothers or fathers) of the same children. The results showed that the responses were similar independent of the caregiver [48]. Therefore, the large number of mothers participating in this study probably did not influence our results.

In this study, the monthly family income was low (about 57.5% received up to 3 minimum wages (MW); Appendix A). The tendency of a low family income was previously mentioned in a study in Brazil that evaluated the quality of life of the parents of children with ASD in which 80.0% of the respondents had a family income up to 4 MW [31]. Our results differed from those in the Brazilian study on FN in neurotypical children, in which 69.1% had a family income higher than 3 MW [25]. The social, personal, and financial impact that the families of children with ASD experience makes living with autism a challenge. It is common for the parents of children with ASD to take care of their children by leaving their jobs (impairing family incomes) [49,50], which justifies the results in this study. Low-income families purchase less fresh fruits and vegetables and more unhealthy foods since they are cheap and energy- and sugar-dense products with a longer shelf-life and are acceptable to children [51]. In this sense, it would be plausible to associate FN with low income due to limited dietary variety. Our results did not show a significant difference in FN with different family income levels (Table 3). This result was similar to a study on neurotypical children [52] and another performed with young adults with ASD in which neophobia was not associated with family income [53]. In our study, most children had only ASD (*n* = 492; 83.0%) and 17.0% of the children with ASD also presented other medical diagnoses (such as food allergies/intolerance, Down syndrome, and others). This result for other medical diagnoses is lower than that in Brazilian children without ASD; 20.1% presented one or more diagnoses [42].

In our study, children with ASD were more neophobic than neurotypical Brazilian children [42]. A similar result was found in children 8 to 11 years old from England and Wales, in which children with ASD were more neophobic than children without ASD [4]. These results were expected since FN is often described as being chronic in ASD because of the insistence on rigid routines during mealtimes. Children with ASD demand food prepared the same way at each presentation and eat in the same ritualistic or obsessive manner at every presentation. Food refusal is commonly based on texture, color, brands and packaging, smell, food group, and temperature [26]. A meta-analysis showed that eating problems among children with ASD are five times higher than in their peers without ASD, suggesting that food selectivity contributes to nutritional impairments in children with ASD [54]. Similarly, our study showed high FN prevalence in 73.9% of the children with ASD, 2.2 times higher than in neurotypical Brazilian children (33.4%) [42]. In England, a study evaluated FN in 4564 children (8–11 years old) from a large community-based sample and a relatively small group of children with ASD (*n* = 37) [4]. Despite fewer participants with ASD, FN was higher in the ASD group than the non-ASD group [4]. In that study, gender and age were significantly associated with FN (*p* < 0.050), and younger children and males had higher FN scores. However, it was impossible to evaluate FN among children with ASD due to the small ASD sample [4]. Unlike our study, in a Brazilian study on FN in children, boys were more neophobic, considering general and fruit neophobia [42]. There was no difference between genders in vegetable FN [42], as in our results. In addition, similarly to our study, the authors did not find differences in FN among ages [42]. FN between age groups was not different in other studies [55,56]. However, studies have found that older children have lower scores on FN scales than younger ones [41,57,58]. As children grow up, FN tends to decrease because of different experiences as they learn about new foods and interact more with their environment [21]. Considering ASD characteristics, it would be expected that FN is most influenced by repetitive behavior than by age and gender, as found in our study.

Although the fruit neophobia domain was relatively low compared to the other domains, there was still a high prevalence of fruit neophobia. Fruits and vegetables are the most rejected food groups by children with ASD, which is associated with inadequate micronutrient intake [26]. This scenario probably occurs with children with ASD included in our study, given the high prevalence of FN of 73.9% in total (73.9% for vegetables, 63.7% for fruits, and 75.7% for the general domain). These results demonstrate that these children are neophobic in various contexts and tend to have a restricted diet. A study performed in the United States with 65 adolescents/young adults with ASD (12–28 years old) and 59 neurotypical adolescents/young adults (12–23 years old) showed that the ASD group was more neophobic than the non-ASD group [59].

The strength of our study was the sample size (*n* = 593) from all regions over the country since it dealt with individuals with specific conditions, such as children with ASD. However, there are some limitations, such as the potential selection bias. In the participants’ selection, we did not use a random sample, making it difficult to generalize the results, particularly those related to the respondent profile (e.g., mother or father) and internet access. The snowball sampling technique may not be the best option for this population, but a large number of participants allowed validation of the results. This method was chosen because it is less costly and less invasive, requiring less effort and time for the researchers and participants than a face-to-face interview. Face-to-face interviews were not possible for this study to reach participants due to geographical limitations [60,61] and the COVID-19 pandemic during data collection. It is important to highlight that data from Brazil showed that three in four Brazilians have internet access. About 93% of adults have a cellphone, the primary internet access tool [62]. In addition, our previous study compared responses given by mothers and fathers using the same questionnaire and showed that their answers were similar independent of being a mother or father [48]. Therefore, the large number of mothers participating in this study probably did not influence our results, but we cannot generalize them. Therefore, the method was considered efficient for data collection during the pandemic, despite the unrepresentativeness of the sample.

## 5. Conclusions

A high prevalence of total FN and others domains of FN was identified in children with ASD from Brazil, confirming that FN is a common problem for children with ASD and deserves more attention. FN did not decrease in older children with ASD, and no association was found with the educational level of caregivers, the number of residents in the same house, and family income. Boys and girls had similar neophobia, which shows that FN is a more complex problem, requiring a multidisciplinary team trained to face the problem.

## Figures and Tables

**Figure 1 children-09-01907-f001:**
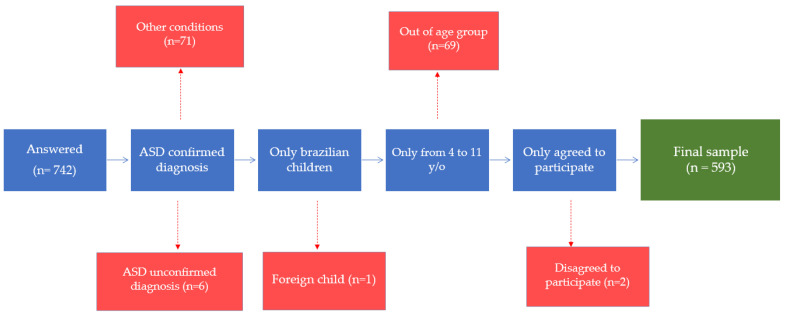
Schematic representation of the study sample (Brazil, 2020–2021). ASD: autistic spectrum disorder.

**Table 1 children-09-01907-t001:** Distribution of the sample according to food neophobia classification in children with ASD (*n* = 593; Brazil, 2020–2021).

	Food Neophobia		
	Low	Moderate	High
General neophobia domain *	39 (6.6%)	105 (17.7%)	449 (75.7%)
Fruit neophobia domain *	68 (11.5%)	147 (24.8%)	378 (63.7%)
Vegetable neophobia domain *	50 (8.4%)	105 (17.7%)	438 (73.9%)
Total score **	50 (8.4%)	107 (18.0%)	436 (73.9%)

* Domain score cut-off points: low—up to 13 points; moderate—from 14 to 21 points; high—22 points or more. ** Total score cut-off points: low—up to 40 points; moderate—from 41 to 65 points; high—66 points or more.

**Table 2 children-09-01907-t002:** Scores and distribution of the food neophobia classification by gender and age group in children with ASD (Brazil, 2020–2021).

	Gender			Age		
	Girls (*n* = 118)	Boys (*n* = 475)	*p*-Value	4–7 Years (*n* = 378)	8–11 Years (*n* = 215)	*p*-Value
General neophobia						
Mean (SD)	24.86 (7.56)	25.84 (7.03)	0.184 *	25.31 (7.18)	26.23 (7.06)	0.135 *
Low (≤13)	10 (8.5%)	29 (6.1%)		27 (7.1%)	12 (5.6%)	
Moderate (14 to 21)	22 (18.6%)	83 (17.5%)	0.596 **	71 (18.8%)	34 (15.8)	0.458 **
High (≥22)	86 (72.9%)	363 (76.4%)		280 (74.1%)	169 (78.6%)	
Fruit neophobia						
Mean (SD)	22.23 (7.63)	23.08 (7.11)	0.252 *	22.61 (7.26)	23.43 (7.14)	0.184 *
Low (≤13)	17 (14.4%)	51 (10.7%)		46 (12.2%)	22 (10.2%)	
Moderate (14 to 21)	27 (22.9%)	120 (25.3%)	0.509 **	87 (23.0%)	60 (27.9%)	0.376 **
High (≥22)	74 (62.7%)	304 (64.0%)		245 (64.8%)	133 (61.9%)	
Vegetable neophobia						
Mean (SD)	24.42 (8.06)	24.87 (6.92)	0.543 *	24.59 (7.16)	25.13 (7.15)	0.379 *
Low (≤13)	13 (11.0%)	37 (7.8%)		33 (8.7%)	17 (7.9%)	
Moderate (14 to 21)	21 (17.8%)	84 (71.2%)	0.519 **	63 (16.7%)	42 (19.5%)	0.662 **
High (≥22)	84 (71.2%)	354 (74.5%)		282 (74.6%)	156 (72.6%)	
Total						
Mean (SD)	71.52 (21.37)	73.79 (19.35)	0.264 *	72.52 (19.94)	74.79 (19.43)	0.179 *
Low (up to 40)	13 (11.0%)	37 (7.8%)		34 (9.0%)	16 (7.4%)	
Moderate (41 to 65)	20 (16.9%)	87 (18.3%)	0.519 **	70 (18.5%)	37 (17.2%)	0.714 **
High (66 or more)	85 (72.0%)	351 (73.9%)		274 (72.5%)	162 (75.3%)	

* Independent *t*-test; ** Pearson’s chi-square test.

**Table 3 children-09-01907-t003:** Distribution of the food neophobia classification by the educational level of caregivers, the number of residents in the same house, and family income (Brazil, 2020–2021).

	Educational Level of Caregivers			Number of Residents in the Same House			Family Income			
	High School (*n* = 314)	Higher Education (*n* = 279)	*p*-Value ****	Up to 3 People (*n* = 140)	4 or More (*n* = 326)	*p*-Value ****	Up to 4 MW *** (*n* = 376)	5 to 9 MW *** (*n* = 105)	10 MW *** or More (*n* = 77)	*p*-Value ****
General neophobia *										
Low	19 (6.1%)	20 (7.2%)		19 (7.3%)	20 (6.0%)		28 (7.4%)	2 (1.9%)	7 (9.1%)	
Moderate	64 (20.3%)	41 (14.7%)	0.184	44 (16.7%)	61 (18.5%)	0.760	71 (18.9%)	17 (16.2%)	11 (14.3%)	0.178
High	231 (73.6%)	218 (78.1%)		199 (76.0%)	250 (75.5%)		277 (73.7%)	86 (81.9%)	59 (76.6%)	
Fruit neophobia *										
Low	34 (10.8%)	34 (12.2%)		33 (12.6%)	35 (10.6%)		45 (12.0%)	11 (10.5%)	8 (10.4%)	
Moderate	78 (24.9%)	69 (24.7%)	0.872	60 (22.9%)	87 (26.3%)	0.538	94 (25.0%)	24 (22.8%)	21 (27.3%)	0.939
High	202 (64.3%)	176 (63.1%)		169 (64.5%)	209 (63.1%)		237 (63.0%)	70 (66.7%)	48 (62.3%)	
Vegetable neophobia *										
Low	21 (6.7%)	29 (10.4%)		22 (8.4%)	28 (8.5%)		32 (8.5%)	7 (6.7%)	10 (13.0%)	
Moderate	53 (16.9%)	52 (18.6%)	0.196	42 (16.0%)	63 (19.0%)	0.628	61 (16.2%)	21 (20.0%)	16 (20.8%)	0.403
High	240 (76.4%)	198 (71.0%)		198 (75.6%)	240 (72.5%)		283 (75.3%)	77 (73.3%)	51 (66.2%)	
Total **										
Low	23 (7.3%)	27 (9.7%)		24 (9.2%)	26 (7.9%)		32 (8.5%)	8 (7.6%)	7 (9.1%)	
Moderate	59 (18.8%)	48 (17.2%)	0.552	44 (16.8%)	63 (19.0%)	0.698	74 (19.7%)	14 (13.4%)	15 (19.5%)	0.629
High	232 (73.9%)	204 (73.1%)		194 (74.0%)	242 (73.1%)		270 (71.8%)	83 (79.0%)	55 (71.4%)	

* Domain score cut-off points: low—up to 13 points; moderate—from 14 to 21 points; high—22 points or more. ** Total score cut-off points: low—up to 40 points; moderate—from 41 to 65 points; high—66 points or more. *** Minimum wage. **** Pearson’s chi-square test.

## Data Availability

Not applicable.

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
