# Peer review of "Food Neophobia in Children with Autistic Spectrum Disorder (ASD): A Nationwide Study in Brazil"

_children, 2022, doi:10.3390/children9121907_

Round 1

Reviewer 1 Report (Previous Reviewer 1)

I think the article is acceptable in its current form.

Author Response

Thanks a lot for opportunity

Reviewer 2 Report (Previous Reviewer 2)

The comments formulated have been sufficiently taken into account. 

Author Response

Thanks for the opportunity

This manuscript is a resubmission of an earlier submission. The following is a list of the peer review reports and author responses from that submission.

Round 1

Reviewer 1 Report

1.     There are many studies on food neophobia in the literature. What does this study offer us differently? How is it different from previous studies? The reason for the study and its hypothesis should be clearly defined (information on lines 51-54 is not sufficient).

2.     What are the ASD diagnostic criteria of the participants included in the study?

3.     Researchers excluded ncomplete questionnaires (spelling error in the sentence should be corrected).

4.     Who identifies as a Caregiver? Mother or father? This may affect the results.

5.     In the tables, some numerical values are given as two numbers, some as one number, and some as three numbers after the dot. Usage should be standard.

6.     Analysis methods should be detailed. For example, the normality distribution.

7.     Limitations of the study should be added.

Author Response

Please see below the responses to reviewers' comments. In the revised manuscript, all the changes are highlighted using the "Track Changes" function in Microsoft Word. Thank you for the opportunity!

Reviewer #1:

  1. There are many studies on food neophobia in the literature. What does this study offer us differently? How is it different from previous studies? The reason for the study and its   hypothesis   should   be   clearly   deffned   (information   on   lines   51-54   is   not sufficient).

R: We apologize for not being clear, added this information to the manuscript.

  1. What are the ASD diagnostic criteria of the participants included in the study?

R: Thank you for your comment. This information is now inserted in the method, section 2.1.

  1. Researchers excluded incomplete questionnaires (spelling error in the sentence should be corrected).

R: We apologize for our mistake. The typo mistake was corrected in section 2.1.

  1. Who identifies as a Caregiver? Mother or father? This may affect the results.

R: Thank you for your comment. This information was inserted in the supplementary file, and now it is also described in the manuscript.

  1. In the tables, some numerical values are given as two numbers, some as one number, and some as three numbers after the dot. Usage should be standard.

R: We apologize for not having noticed the error in some data. We standard numerical values in one number in percentages, means and standard deviation with two numbers, and only p value with three numbers. We also fixed some typos in the tables. We took the opportunity to put the title of Figure 1 in the text, not in the image as it was before.

  1. Analysis methods should be detailed. For example, the normality distribution.

R: Thank you for your comment. We include the description of the test for checking the assumption of normality.

  1. Limitations of the study should be added

R: Thank you for your comment. The limitations were added in the last paragraph of the discussion section.

Thank you for your comments and the opportunity to improve our manuscript!

Reviewer 2 Report

The issue of feeding children with autism is one of the frequently addressed. The authors are well versed in the topic of the article (I appreciate the accurate references to many studies). 

The introduction lacked more detailed information about food neophobia - types, conditions. This is worth completing.

The research design is correct. The method was well chosen. The selection of subjects by snowball recruitment method is questionable, but the relatively large group of subjects allows cautious generalization.  The results collected are interesting. The discussion of the results is correct, and the conclusions are valid. 

Author Response

Please see below the responses to reviewers’ comments. In the revised manuscript, all the changes are highlighted using the “track changes” function in Microsoft Word. Thank you for the opportunity to improve our manuscript!

Comments and Suggestions for Authors

Reviewer #2:

  1. The issue of feeding children with autism is one of the frequently addressed. The authors are well versed in the topic of the article (I appreciate the accurate references to many studies).

R: We appreciate your comment, we strive to bring what is important and relevant references, as well as the most up-to-date references.

  1. The introduction lacked more detailed information about food neophobia - types, conditions. This is worth completing.

R: Thanks so much for the suggestion. In lines 47 to 52 we add the scenarios in which food neophobia is investigated. Its main types in childhood are in lines 56 and 57, as well as the consumption of substitute foods.

  1. The research design is correct. The method was well chosen. The selection of subjects by snowball recruitment method is questionable, but the relatively large group of subjects allows cautious generalization. The results collected are interesting. The discussion of the results is correct, and the conclusions are valid.

R: Thank you for your comment. We complemented the limitations of our study by adding information from the snowball method not being the best option in lines 259 and 260.

Thank you for the opportunity to improve our manuscript!

Round 2

Reviewer 1 Report

1.     I saw that the authors made an arrangement in the author name order. What is the reason for this and was consent obtained from all authors in this study?

2.     ASD diagnostic criteria (Exp. DSM V????) information regarding the participants in the study is still lacking and this part should be clarified.

3.     The use of numbers after the dot in the tables varies. This part should be done again, taking into account previous revisions.

4.     The study was completed with 593 participants. Why was the Shapiro-Wilk test preferred for normality distribution?

5.     I think that the change of caregivers (eg mother, father) is a limitation. This should be added to the limitations.

Author Response

Please see below the responses to reviewers’ comments. In the revised manuscript, all the changes are highlighted using the “track changes” function in Microsoft Word. Thank you for the opportunity to improve our manuscript!

Comments and Suggestions for Authors

Reviewer #1:

  1. I saw that the authors made an arrangement in the author name order. What is the reason for this and was consent obtained from all authors in this study?

R: Thank you for your comment. We apologize for this inconvenience, but we just realized the mistake in the author names order after submission. The order was organized based on author’s contributions. All authors agree and consent to the name order.

  1. ASD diagnostic criteria (Exp. DSM V????) information regarding the participants in the study is still lacking and this part should be clarified.

R: According to the DSM-5, Autism Spectrum Disorder encompasses autism, Asperger's, Rett, childhood disintegrative disorder, pervasive developmental disorder, and is characterized by deficits in two important domains, the first is the deficit in social communication and social interaction, and the second the presence of restricted and repetitive patterns of behavior, interests, and activities.

  1. The use of numbers after the dot in the tables varies. This part should be done again, taking into account previous revisions.

R: We apologize for our mistake. It was corrected.

  1. The study was completed with 593 participants. Why was the Shapiro-Wilk test preferred for normality distribution?

R: In their seminal paper (Shapiro, S. S.; Wilk, M. B. (1965). "An analysis of variance test for normality (complete samples)". Biometrika. 52 (3–4): 591–611. doi:10.1093/biomet/52.3-4.591 ) they only simulated data with a maximum N of 50. Thus, they seemed to focus on improving the testing power for small sample sizes. Although textbooks in statistics suggest using the KS test (with lilliefors adjustment) for large samples, the S-W test can also be used for large samples. In any case, the scores were subjected to the KS test (this information has been updated in the manuscript) and the results were, in essence, similar.

  1. I think that the change of caregivers (eg mother, father) is a limitation. This should be added to the limitations.

R: Thank you for your comment. It was inserted at the end of the discussion section as a limitation of our study.
